# Breaking the Curse of Multiagency in
# Robust Multi-Agent Reinforcement Learning

**Laixi Shi** [* 1]   **Jingchu Gai** [* 2]   **Eric Mazumdar** [3]   **Yuejie Chi** [4]   **Adam Wierman** [3]

## Abstract

Standard multi-agent reinforcement learning (MARL) algorithms are vulnerable to sim-to-real gaps. To address this, distributionally robust Markov games (RMGs) have been proposed to enhance robustness in MARL by optimizing the worst-case performance when game dynamics shift within a prescribed uncertainty set. RMGs remains under-explored, from reasonable problem formulation to the development of sample-efficient algorithms. Two notorious and open challenges are the formulation of the uncertainty set and whether the corresponding RMGs can overcome the curse of multiagency, where the sample complexity scales exponentially with the number of agents. In this work, we propose a natural class of RMGs inspired by behavioral economics, where each agent's uncertainty set is shaped by both the environment and the integrated behavior of other agents. We first establish the well-posedness of this class of RMGs by proving the existence of game-theoretic solutions such as robust Nash equilibria and coarse correlated equilibria (CCE). Assuming access to a generative model, we then introduce a sample-efficient algorithm for learning the CCE whose sample complexity scales polynomially with all relevant parameters. To the best of our knowledge, this is the first algorithm to break the curse of multiagency for RMGs, regardless of the uncertainty set formulation.

---

*Equal contribution [1]Department of Electrical and Computer Engineering, Johns Hopkins University, MD, USA [2]Machine Learning Department, Carnegie Mellon University, PA, USA [3]Department of Computing Mathematical Sciences, California Institute of Technology, CA, USA [4]Department of Statistics and Data Science, Yale University, CT, USA. Correspondence to: Laixi Shi <shilaixi@gmail.com>.

*Proceedings of the 42$^{nd}$ International Conference on Machine Learning*, Vancouver, Canada. PMLR 267, 2025. Copyright 2025 by the author(s).

## 1 Introduction

A flurry of problems naturally involve decision-making among multiple players, whether human, artificial intelligence, or both, with strategic objectives. Multi-agent reinforcement learning (MARL) serves as a powerful framework to address these challenges, demonstrating potential in various applications such as social dilemmas (Leibo et al., 2017; Baker, 2020; Zhang et al., 2024), autonomous driving (Lillicrap et al., 2015), robotics (Kober et al., 2013; Rusu et al., 2017), and games (Mnih et al., 2015; Vinyals et al., 2019). Despite the recent success of standard MARL, its transition from prototypes to reliable production is hindered by robustness concerns due to the complexity and variability of both the real-world environment and human behaviors. Specifically, environmental uncertainty can arise from sim-to-real gaps (Tobin et al., 2017), unexpected disturbance (Pinto et al., 2017), system noise, and adversarial attacks (Mahmood et al., 2018); agents' behaviors are subject to unknown bounded rationality and variability (Tversky & Kahneman, 1974). The solution learned at training can fail catastrophically when faced with a slightly shifted MARL problem during deployment, resulting in a significant drop in overall outcomes and each agent's individual payoff (Balaji et al., 2019; Zhang et al., 2020a; Zeng et al., 2022; Yeh et al., 2021; Shi et al., 2024b; Slumbers et al., 2023).

To address robustness challenges, a promising and flexible framework is (distributionally) robust Markov games (RMGs) (Littman, 1994; Shapley, 1953). It is a robust counterpart to the common playground of standard MARL problems — Markov games (MGs) (Zhang et al., 2020c; Kardeş et al., 2011). In standard MGs, agents consider (competitive) personal objectives and simultaneously interact with each other within a shared unknown environment. The goal is to learn some rationally optimal solution concepts called equilibria, which are joint strategies/policies of agents that all of them stick with rationally with other agents fixed; for instance, Nash equilibria (NE) (Nash, 1951; Shapley, 1953), correlated equilibria (CE), and coarse correlated equilibria (CCE) (Aumann, 1987; Moulin & Vial, 1978). To promote robustness, RMGs differ from standard MGs by defining each agent's payoff (objective) as its worst-case performance when the dynamics of the game shift within a

prescribed uncertainty set centered around a nominal environment.

## 1.1 Open challenges of robust MARL

**Construction of realistic uncertainty sets.** The family of RMGs is a rich class of problems because of the flexibility in constructing the uncertainty sets to capture different uncertainty considerations. The uncertainty sets prevalent in current approaches are constructed under the $(s, \boldsymbol{a})$-*rectangularity condition*, yielding each agent's objective as the expectation over the independent risk-aware outcome on each joint action of other agents' strategies. While observations from behavioral economics (Friedman & Mauersberger, 2022; Sandomirskiy et al., 2024; Goeree et al., 2005; Mazumdar et al., 2024) reveal that, to handle other players' uncertainty, people often use a risk-aware metric outside of the expected outcome of other players' joint policy, rather than flipping the expectation and the risk metric as that of $(s, \boldsymbol{a})$-*rectangularity condition*. To account for realistic human decision-making, we are motivated to develop new classes of RMGs that foster robust solutions for practical MARL problems.

**The curse of multiagency.** Sample efficiency is a crucial challenge for solving MARL due to the limited availability of data relative to the high dimensionality of the problem. In MARL, agents strive to learn through interactions with an unknown environment (Silver et al., 2016; Vinyals et al., 2019; Achiam et al., 2023; Yang et al., 2025) that is often extremely large-scale, while data acquisition can be prohibitively limited by high costs and stakes. As such, a notable scalability challenge is the *the curse of multiagency* — the sample complexity requirement scales exponentially with the number of agents (induced by the exponentially growing size of the joint action space). This issue has been recognized and studied in extensive MARL problems (Song et al., 2021; Rubinstein, 2017), but remains unresolved for robust MARL. We concentrate on finite-horizon multiplayer general-sum Markov games, with a widely-used data collection mechanism — generative model (Kearns & Singh, 1999), where the number of agents is $n$, the episode length is $H$, the size of the state space is $S$, and the size of the $i$-th agent's action space is $A_i$, for $1 \le i \le n$.

- *Breaking the curse of multiagency in standard MARL.* A line of pioneering work (Jin et al., 2021; Bai & Jin, 2020; Song et al., 2021; Li et al., 2023) has recently introduced a new suite of algorithms using adaptive sampling that provably break the curse of multiagency in standard MGs. In particular, to find an $\varepsilon$-approximate CCE, Li et al. (2023) requires a minimax-optimal sam-

ple complexity no more than

$$\widetilde{O}\left(\frac{H^4 S \sum_{i=1}^n A_i}{\varepsilon^2}\right) \quad (1)$$

up to logarithmic factors, which depends only on the sum of individual actions, rather than the number of joint actions.

- *The persistent curse of multiagency in robust MARL.* The development of provable sample-efficient algorithms for RMGs is largely underexplored, with only a few recent studies (Zhang et al., 2020c; Kardeş et al., 2011; Ma et al., 2023; Blanchet et al., 2023; Shi et al., 2024b). Focusing on a class of RMGs with uncertainty sets satisfying the $(s, \boldsymbol{a})$-*rectangularity condition*, existing works all suffer from the curse of multiagency, significantly limiting their scalability. For example, using the total variation (TV) distance as the divergence function, Shi et al. (2024b) relying on non-adaptive sampling, finds an $\varepsilon$-approximate robust CCE with a sample complexity no more than

$$\widetilde{O}\left(\frac{H^3 S \prod_{i=1}^n A_i}{\varepsilon^2} \min\left\{H, \frac{1}{\min_{1\le i \le n}\sigma_i}\right\}\right) \quad (2)$$

up to logarithmic factors, where $\sigma_i \in [0, 1)$ is the uncertainty level for the $i$-th agent. As a result, the sample size requirement becomes prohibitive when the number of agents is large. Consequently, there is a significant desire to explore paths that could break through the curse of multiagency in RMGs, which is much more involved than its standard counterpart due to complicated non-linearity introduced by planning for worst-case performances.

Given these two challenges of uncertainty set construction and the curse of multiagency, it raises an open question:

*Can we design RMGs with realistic uncertainty sets that come with sample complexity guarantees breaking the curse of multiagency?*

## 1.2 Contributions

Inspired by behavioral economics, we propose a new class of RMGs with a *fictitious* uncertainty set that explicitly models environmental uncertainties from the perspective of realistic human players, making it suitable for complex real-world scenarios. We begin by verifying the game-theoretic properties of the proposed class of RMGs to ensure the existence of robust variants of well-known standard equilibria notions, robust NE and robust CCE. Next, due to the general intractability of learning NE, we focus on designing algorithms that can provably overcome the curse of multiagency in learning an approximate robust CCE, referring to

| Algorithm | Uncertainty set | Equilibria | Sample complexity |
|---|---|---|---|
| P$^2$MPO (Blanchet et al., 2024) | $(s, \boldsymbol{a})$-rectangularity | robust NE | $S^4 \left(\prod_{i=1}^n A_i\right)^3 H^4/\varepsilon^2$ |
| DR-NVI (Shi et al., 2024b) | $(s, \boldsymbol{a})$-rectangularity | robust NE/CE/CCE | $\frac{SH^3 \prod_{i=1}^n A_i}{\varepsilon^2} \min\left\{H, \frac{1}{\min_{1\leq i\leq n}\sigma_i}\right\}$ |
| Robust-Q-FTRL **(this work)** | fictitious $(s, a_i)$-rectangularity | robust CCE | $\frac{SH^6 \sum_{1\leq i\leq n} A_i}{\varepsilon^4} \min\left\{H, \frac{1}{\min_{1\leq i\leq n}\sigma_i}\right\}$ |

*Table 1.* We show all the existing sample complexity results within the general context of robust Markov games (RMGs) to put our work into perspective, on finding an $\varepsilon$-approximate equilibrium in finite-horizon multi-agent general-sum robust MG, omitting logarithmic factors. Our result is the only algorithm that breaks the curse of multiagency regardless of the RMG formulations.

a joint policy where no agent can improve their benefit by more than $\varepsilon$ through rational deviations.. Specifically, for sampling mechanisms to explore the unknown environment, we assume access to a generative model that can only draw samples from the nominal environment (Shi et al., 2024b). The main contributions are summarized as follows.

- We introduce a new class of robust Markov games using fictitious uncertainty sets with *others-integrated* $(s, a_i)$-*rectangularity condition* (see Section 2.2 for details), which is not only realistic viewpoint observed from behavioral economics, but also a natural adaptation from robust single-agent RL to robust MARL. The uncertainty set for each agent $i$ can be decomposed into independent subsets over each state and its own action tuple $(s, a_i)$, where each subset is a "ball" around the expected nominal transition determined by other agents' policies and the nominal transition kernel, a distance function $\rho$, and the radius/uncertainty level $\sigma_i$. We verify several essential facts of this class of RMGs: the existence of the desired equilibrium — robust NE and robust CCE for this new class of RMGs using game-theoretical tools such as fixed-point theorem; the existence of best-response policies and robust Bellman equations.

- We consider the total variation (TV) distance as the distance metric $\rho$ for uncertainty sets due to its popularity in both theory (Panaganti & Kalathil, 2022; Shi et al., 2023; Blanchet et al., 2023; Shi et al., 2024b) and practice (Pan et al., 2023; Lee et al., 2021; Szita et al., 2003). Focusing on the proposed RMGs with fictitious uncertainty sets, we design Robust-Q-FTRL that can provably find $\varepsilon$-approximate robust CCE with high probability, as long as the sample size exceeds

$$\widetilde{O}\left(\frac{SH^6 \sum_{i=1}^n A_i}{\varepsilon^4} \min\left\{H, \frac{1}{\min_{1\leq i\leq n}\sigma_i}\right\}\right) \quad (3)$$

up to logarithmic factors, where $\sigma_i \in (0, 1]$ is the uncertainty level for the $i$-th agent. To the best of

our knowledge, this is the first algorithm to break the curse of multiagency in sample complexity of RMGs regardless of the uncertainty set definition, which can provably find an $\varepsilon$-approximate robust CCE using a sample size that is polynomial to all salient parameters. Table 1 provides a detailed summary of all existing sample complexity results in robust MARL[1], where our results show significantly data efficiency with linear dependency on the size of each agent's action space, which is absent from prior works (Blanchet et al., 2024; Shi et al., 2024b). To achieve this, we utilize adaptive sampling and online adversarial learning tools, coupled by a tailored design and analysis for robust MARL due to the nonlinearity of the robust value function, which contrasts with the linear payoff functions in standard MARL with respect to the transition kernel.

**Notation.** In this paper, we denote $[T] := \{1, 2, \ldots, T\}$ for any positive integer $T > 0$. We define $\Delta(\mathcal{S})$ as the simplex over a set $\mathcal{S}$. For any policy $\pi$ and function $Q(\cdot)$ defined over a domain $\mathcal{B}$, the variance of $Q$ under $\pi$ is given by $\mathsf{Var}_\pi(Q) := \sum_{a\in\mathcal{B}} \pi(a)[Q(a) - \mathbb{E}_\pi[Q]]^2$. We define $x = [x(s, \mathbf{a})]_{(s,\mathbf{a})\in\mathcal{S}\times\mathcal{A}} \in \mathbb{R}^{SA}$ as any vector that represents values for each state-action pair, and $x = [x(s, a_i)]_{(s,a_i)\in\mathcal{S}\times\mathcal{A}_i} \in \mathbb{R}^{SA_i}$ as any vector representing agent-wise state-action values. Similarly, we denote $x = [x(s)]_{s\in\mathcal{S}}$ as any vector representing values for each state. For $\mathcal{X} := (\mathcal{S}, \{A_i\}_{i\in[n]}, H, \{\sigma_i\}_{i\in[n]}, \frac{1}{\varepsilon}, \frac{1}{\delta})$, let $f(\mathcal{X}) = O(g(\mathcal{X}))$ denote that there exists a universal constant $C_1 > 0$ such that $f \leq C_1 g$. Furthermore, the notation $\widetilde{O}(\cdot)$ is defined similarly to $O(\cdot)$ but hides logarithmic factors.

---

[1]Note that, since we focus on a new class of RMGs, the sample complexity results in this work cannot be directly compared to those in prior studies. However, we provide a summary in Table 1 of existing sample complexity results for general RMGs, regardless of the uncertainty set formulation, for reference.

## 2 Preliminaries

In this section, we begin with some background on multi-agent general-sum standard Markov games (MGs) in finite-horizon settings, followed by a general framework of a robust variant of standard MGs —- distributionally robust Markov games.

### 2.1 Standard Markov games

A finite-horizon *multi-agent general-sum Markov game* (MG) can be characterized by the tuple

$$\mathcal{MG} = \big\{ \mathcal{S}, \{\mathcal{A}_i\}_{1\le i \le n}, P, r, H \big\}.$$

This setup features $n$ agents each striving to maximize their individual long-term cumulative rewards within a shared environment. At each time step, all agents observe the same state over the state space $\mathcal{S} = \{1, \cdots, S\}$ within the shared environment. For each agent $i$ ($i \in [n]$), $\mathcal{A}_i = \{1, \cdots, A_i\}$ denotes its action space containing $A_i$ possible actions. The joint action space for all agents (resp. the subset excluding the $i$-th agent) is defined as $\mathcal{A} := \mathcal{A}_1 \times \cdots \times \mathcal{A}_n$ (resp. $\mathcal{A}_{-i} := \prod_{j\ne i} \mathcal{A}_j$ for any $i \in [n]$). We use the notation $\boldsymbol{a} \in \mathcal{A}$ (resp. $\boldsymbol{a}_{-i} \in \mathcal{A}_{-i}$) to denote a joint action profile involving all agents (resp. all except the $i$-th agent). In addition, the probability transition kernel $P = \{P_h\}_{1\le h \le H}$, with each $P_h : \mathcal{S} \times \mathcal{A} \mapsto \Delta(\mathcal{S})$, describes the dynamics of the game: $P_h(s' \mid s, \boldsymbol{a})$ is the probability of transitioning from state $s \in \mathcal{S}$ to state $s' \in \mathcal{S}$ at time step $h$ when agents choose the joint action profile $\boldsymbol{a} \in \mathcal{A}$. The reward function of the game is $r = \{r_{i,h}\}_{1\le i \le n, 1 \le h \le H}$, with each $r_{i,h} : \mathcal{S} \times \mathcal{A} \mapsto [0,1]$ normalized to the unit interval. For any $(i, h, s, \boldsymbol{a}) \in [n] \times [H] \times \mathcal{S} \times \mathcal{A}$, $r_{i,h}(s, \boldsymbol{a})$ represents the immediate reward received by the $i$-th agent in state $s$ when the joint action profile $\boldsymbol{a}$ is taken. Lastly, $H > 0$ represents the horizon length.

**Markov policies and value functions.** In this work, we concentrate on Markov policies that the action selection rule depends only on the current state $s$, independent from previous trajectory. Namely, the $i$-th ($i \in [n]$) agent chooses actions according to $\pi_i = \{\pi_{i,h} : \mathcal{S} \mapsto \Delta(\mathcal{A}_i)\}_{1\le h \le H}$. Here, $\pi_{i,h}(a \mid s)$ represents the probability of selecting action $a \in \mathcal{A}_i$ in state $s$ at time step $h$. As such, the joint Markov policy of all agents can be denoted as $\pi = (\pi_1, \ldots, \pi_n) : \mathcal{S} \times [H] \mapsto \Delta(\mathcal{A})$, i.e., given any $s \in \mathcal{S}$ and $h \in [H]$, the joint action profile $\boldsymbol{a} \in \mathcal{A}$ of all agents is chosen following the distribution $\pi_h(\cdot \mid s) = (\pi_{1,h}, \pi_{2,h} \ldots, \pi_{n,h})(\cdot \mid s) \in \Delta(\mathcal{A})$.

To continue, for any given joint policy $\pi$ and transition kernel $P$ of a $\mathcal{MG}$, the $i$-th agent's long-term cumulative reward can be characterized by the value function $V_{i,h}^{\pi,P} : \mathcal{S} \mapsto \mathbb{R}$ (resp. Q-function $Q_{i,h}^{\pi,P} : \mathcal{S} \times \mathcal{A} \mapsto \mathbb{R}$) as below:

for all $(h, s, a) \in [H] \times \mathcal{S} \times \mathcal{A}$,

$$V_{i,h}^{\pi,P}(s) := \mathbb{E}_{\pi,P}\left[ \sum_{t=h}^{H} r_{i,t}(s_t, \boldsymbol{a}_t) \mid s_h = s \right],$$

$$Q_{i,h}^{\pi,P}(s, \boldsymbol{a}) := \mathbb{E}_{\pi,P}\left[ \sum_{t=h}^{H} r_{i,t}(s_t, \boldsymbol{a}_t) \mid s_h = s, \boldsymbol{a}_h = \boldsymbol{a} \right].$$
$$(4)$$

In this context, the expectation is calculated over the trajectory $\{(s_t, \boldsymbol{a}_t)\}_{h \le t \le H}$ produced by following the joint policy $\pi$ under the transition kernel $P$.

### 2.2 Distributionally robust Markov games

A general distributionally robust Markov game (RMG) is represented by the tuple

$$\mathcal{RMG} = \big\{ \mathcal{S}, \{\mathcal{A}_i\}_{1\le i \le n}, \{\mathcal{U}_\rho^{\sigma_i}(P^0, \cdot)\}_{1\le i \le n}, r, H \big\}.$$

Here, $\mathcal{S}, \{\mathcal{A}_i\}_{1\le i \le n}, r, H$ are defined in the same manner as those in standard MGs (see Section 2.1). RMGs differ from standard MGs: for each agent $i$ ($1 \le i \le n$), the transition kernel is not fixed but can vary within its own prescribed uncertainty set $\mathcal{U}_\rho^{\sigma_i}(P^0, \cdot)$ determined by (possibly the current policy and) a *nominal* kernel $P^0 : H \times \mathcal{S} \times \mathcal{A} \mapsto \Delta(\mathcal{S})$ that represents a reference (such as the training environment). The shape and the size of the uncertainty set $\big\{\mathcal{U}_\rho^{\sigma_i}(P^0, \cdot)\big\}_{i\in[n]}$ are further specified by a divergence function $\rho$ and the uncertainty levels $\{\sigma_i\}_{i\in[n]}$, serving as the "distance" metric and the radius respectively.

Various choices of the divergence function have been considered in the literature of robust RL, including but not limited to $f$-divergence (such as total variation, $\chi^2$ divergence, and Kullback-Leibler (KL) divergence) (Yang et al., 2022; Zhou et al., 2021; Shi & Chi, 2024; Lu et al., 2024; Wang et al., 2024) and Wasserstein distance (Xu et al., 2023). Adopting uncertainty sets with different structures leads to distinct RMGs, as they address distinct types of uncertainty and game-theoretical solutions. This paper focuses on variability in environmental dynamics (transition kernels), though uncertainty in agents' reward functions could also be considered similarly but is omitted for brevity.

**Robust value functions and best-response policies.** For any RMG, each agent seeks to maximize its worst-case performance in the presence of other agents' behaviors despite perturbations in the environment dynamics, as long as the kernel transitions remain within its prescribed uncertainty set. Mathematically, given any joint policy $\pi : \mathcal{S} \times [H] \mapsto \Delta(\mathcal{A})$, the worst-case performance of any agent $i$ is characterized by the *robust value function* $V_{i,h}^{\pi,\sigma_i}$ and the *robust Q-function* $Q_{i,h}^{\pi,\sigma_i}$: for all $(i, h, s, a_i) \in [n] \times [H] \times \mathcal{S} \times \mathcal{A}_i$,

$$V_{i,h}^{\pi,\sigma_i}(s) := \inf_{P \in \mathcal{U}_\rho^{\sigma_i}(P^0, \pi)} V_{i,h}^{\pi,P}(s)$$

$$Q_{i,h}^{\pi,\sigma_i}(s, a_i) := \inf_{P \in \mathcal{U}_\rho^{\sigma_i}(P^0, \pi)} Q_{i,h}^{\pi,P}(s, a_i). \quad (5)$$

Note that different from (4), here the Q-function for any $i$-th agent is defined only over its own action $a_i \in \mathcal{A}_i$ rather than the joint action $\boldsymbol{a} \in \mathcal{A}$.

To continue, we denote $\pi_{-i}$ as the policy for all agents except for the $i$-th agent. By optimizing the $i$-th agent's policy $\pi_i' : \mathcal{S} \times [H] \to \Delta(\mathcal{A}_i)$ (independent from $\pi_{-i}$), we define the maximum of the robust value function as

$$V_{i,h}^{\star,\pi_{-i},\sigma_i}(s) := \max_{\pi_i':\mathcal{S}\times[H]\mapsto\Delta(\mathcal{A}_i)} V_{i,h}^{\pi_i'\times\pi_{-i},\sigma_i}(s)$$

$$= \max_{\pi_i':\mathcal{S}\times[H]\mapsto\Delta(\mathcal{A}_i)} \inf_{P\in\mathcal{U}_\rho^{\sigma_i}(P^0,\pi)} V_{i,h}^{\pi_i'\times\pi_{-i},P}(s) \quad (6)$$

for all $(i, h, s) \in [n] \times [H] \times \mathcal{S}$. The policy that achieves the maximum of the robust value function for all $(i, h, s) \in [n] \times [H] \times \mathcal{S}$ is called a *robust best-response policy*.

**Solution concepts for robust Markov games.** In view of the conflicting objectives between agents, establishing equilibrium becomes the goal of solving RMGs. As such, we introduce two kinds of solution concepts — robust NE and robust CCE — robust variants of standard NE and CCE (usually considered in standard MGs) specified to the form of RMGs.

- *Robust NE.* A product policy $\pi = \pi_1 \times \pi_2 \times \cdots \times \pi_n : \mathcal{S} \times [H] \mapsto \prod_{i=1}^n \Delta(\mathcal{A}_i)$ is said to be a *robust NE* if

$$V_{i,1}^{\pi,\sigma_i}(s) = V_{i,1}^{\star,\pi_{-i},\sigma_i}(s), \quad \forall (s, i) \in \mathcal{S} \times [n]. \quad (7)$$

Given the strategies of the other agents $\pi_{-i}$, when each agent wants to optimize its worst-case performance when the environment and other agents' policy stay within its own uncertainty set $\mathcal{U}_\rho^{\sigma_i}(P^0, \pi)$, robust NE means that no player can benefit by unilaterally diverging from its present strategy.

- *Robust CCE.* A distribution over the joint product policy $\xi := \{\xi_h\}_{h\in[H]} : S \times [H] \mapsto \Delta(\prod_{i\in[n]} \Delta(\mathcal{A}_i))$ is said to be a *robust CCE* if it holds that for all $(i, s) \in [n] \times \mathcal{S}$,

$$\mathbb{E}_{\pi\sim\xi}\left[V_{i,1}^{\pi,\sigma_i}(s)\right] \geq \mathbb{E}_{\pi\sim\xi}\left[V_{i,1}^{\star,\pi_{-i},\sigma_i}(s)\right]. \quad (8)$$

Considering all agents follow the policy drawn from the distribution $\xi$, i.e., $\pi_h(s) \sim \xi_h(s)$ for all $(s, h) \in \mathcal{S} \times [H]$, when the distribution of all agents but the $i$-th agent's policy is fixed as the marginal distribution of $\xi$, robust CCE indicates that no agent can benefit from deviating from its current policy.

Note that, for standard MGs, CCE is defined as a possibly correlated joint policy $\pi^{\mathsf{CCE}} : \mathcal{S} \times [H] \mapsto \Delta(\mathcal{A})$ (Moulin

& Vial, 1978; Aumann, 1987) if it holds that for all $(i, s) \in [n] \times \mathcal{S}$,

$$V_{i,1}^{\pi^{\mathsf{CCE}},P}(s) \geq \max_{\pi_i':\mathcal{S}\times[H]\to\Delta(\mathcal{A}_i)} V_{i,1}^{\pi_i'\times\pi_{-i}^{\mathsf{CCE}},P}(s). \quad (9)$$

This correlated policy $\pi^{\mathsf{CCE}}$ can also be viewed as a distribution $\xi$ over the product policy space since each joint action $\boldsymbol{a}$ can be seen as a deterministic product policy. Careful readers may note that the definition (9) of CCE in standard MGs is in a different form from the one (8) in RMGs, as the latter does not include the expectation operator $\mathbb{E}_{\pi\sim\xi}[\cdot]$ with respect to the policy distribution ($\xi$) over the value function. We emphasize that the definition with the expectation operator outside of the value (or cost) function with respect to a distribution of product pure strategies in (8) is a natural formulation originating from game theory (Moulin et al., 2014; Moulin & Vial, 1978). In standard MARL and previous robust MARL studies, the definition in (9) is typically used because (9) and (8) are identical in those situations, as the expectation operator and the corresponding value functions are linear with respect to the joint policy, allowing them to be interchanged (Li et al., 2023; Shi et al., 2024b).

# 3 Robust Markov Games with Fictitious Uncertainty Sets

Given the definition of general RMGs, a natural question arises: what kinds of uncertainty sets should we consider to achieve the desired robustness in our solutions? To address this, we focus on a class of RMGs characterized by a type of natural yet realistic uncertainty sets inspired from behavioral economics. More discussions of this class of games are provided momentarily.

## 3.1 A novel uncertainty set definition in RMGs

We propose a new class of uncertainty sets, named *fictitious uncertainty sets*, which count in the uncertainty induced by both the environment and other agents' behaviors in an integrated manner. Before introducing the uncertainty sets, we provide some auxiliary notations as below. We denote a vector of any transition kernel $P : \mathcal{S} \times \mathcal{A} \mapsto \Delta(\mathcal{S})$ or $P^0 : \mathcal{S} \times \mathcal{A} \mapsto \Delta(\mathcal{S})$ respectively as: for all $(s, \boldsymbol{a}) \in \mathcal{S} \times \mathcal{A}$,

$$P_{h,s,\boldsymbol{a}} := P_h(\cdot \mid s, \boldsymbol{a}) \in \mathbb{R}^{1\times S},$$
$$P_{h,s,\boldsymbol{a}}^0 := P_h^0(\cdot \mid s, \boldsymbol{a}) \in \mathbb{R}^{1\times S}. \quad (10)$$

For any (possibly correlated) joint Markov policy (defined in Section 2.1) $\pi : \mathcal{S} \times [H] \mapsto \Delta(\mathcal{A})$, we define the expected nominal transition kernel conditioned on the situation that the $i$-th agent chooses some action $a_i \in \mathcal{A}_i$ and other agents play according to the conditional policy (i.e., $\boldsymbol{a}_{-i} \sim \pi_h(\cdot \mid s, a_i)$) given $s \in \mathcal{S}$ and $a_i$ as below: for all $(h, s, a_i) \in [H] \times \mathcal{S} \times \mathcal{A}_i$:

$$P_{h,s,a_i}^{\pi_{-i}} = \mathbb{E}_{\boldsymbol{a}\sim\pi_h(\cdot \mid s, a_i)}\left[P_{h,s,\boldsymbol{a}}^0\right]$$

$$= \sum_{\boldsymbol{a}_{-i} \in \mathcal{A}_{-i}} \frac{\pi_h(a_i, \boldsymbol{a}_{-i} \mid s)}{\pi_{i,h}(a_i \mid s)} \left[ P_{h,s,\boldsymbol{a}}^0 \right]. \qquad (11)$$

Armed with the above definitions, now we are in a position to define the *fictitious* uncertainty sets, which satisfy a *others-integrated $(s, a_i)$-rectangularity condition*.

**Definition 3.1.** For any joint policy $\pi : \mathcal{S} \times [H] \mapsto \Delta(\mathcal{A})$, divergence function $\rho : \Delta(\mathcal{S}) \times \Delta(\mathcal{S}) \mapsto \mathbb{R}^+$ and accessible uncertainty levels $\sigma_i \geq 0$ for all $i \in [n]$, the fictitious uncertainty sets $\{\mathcal{U}_\rho^{\sigma_i}(P^0, \pi)\}_{i \in [n]}$ satisfy the *others-integrated $(s, a_i)$-rectangularity* condition: for all $i \in [n]$ and $(h, s, a_i) \in [H] \times \mathcal{S} \times \mathcal{A}_i$,

$$\mathcal{U}_\rho^{\sigma_i}(P^0, \pi) := \otimes \mathcal{U}_\rho^{\sigma_i}\left( P_{h,s,a_i}^{\pi_{-i}} \right), \text{ s.t.}$$
$$\mathcal{U}_\rho^{\sigma_i}\left( P_{h,s,a_i}^{\pi_{-i}} \right) := \left\{ P \in \Delta(\mathcal{S}) : \rho\left( P, P_{h,s,a_i}^{\pi_{-i}} \right) \leq \sigma_i \right\}, \quad (12)$$

where $\otimes$ represents the Cartesian product.

In words, conditioned on a fixed joint policy $\pi$, the uncertainty set $\mathcal{U}_\rho^{\sigma_i}(P^0, \pi)$ for each $i$-th agent can be decomposed into a Cartesian product of subsets over each state and agent-action pair $(s, a_i)$. Each uncertainty subset $\mathcal{U}_\rho^{\sigma_i}(P_{h,s,a_i}^{\pi_{-i}})$ over $(s, a_i)$ is defined as a "ball" around a reference — the expected nominal transition kernel $P_{h,s,a_i}^{\pi_{-i}}$ conditioned on both transition kernel and agents' joint policy $\pi$.

**Further discussions of fictitious uncertainty set.** Here, we discuss the proposed fictitious uncertainty sets, focusing on their practical implications, properties, and relation to prior works. Prior works on RMGs typically focused on a type of uncertainty sets with $(s, \boldsymbol{a})$-*rectangularity condition* (Ma et al., 2023; Blanchet et al., 2023; Shi et al., 2024b). This class of uncertainty sets decouples the uncertainty into independent subsets for each state-joint action pair $(s, \boldsymbol{a})$, accounting for the uncertainty induced by other agents separately and independently, mathematically defined as

$$\mathcal{U}_\rho^{\sigma_i}(P^0) := \otimes \mathcal{U}^{\sigma_i}(P_{h,s,\boldsymbol{a}}^0), \quad \text{where}$$
$$\mathcal{U}_\rho^{\sigma_i}(P_{h,s,\boldsymbol{a}}^0) = \left\{ P_{h,s,\boldsymbol{a}} \in \Delta(\mathcal{S}) : \rho(P_{h,s,\boldsymbol{a}}, P_{h,s,\boldsymbol{a}}^0) \leq \sigma_i \right\}.$$

- *Realistic and predictive of human decisions in comparisons to prior works.* Observed from experimental data of behavioral economics, in many games considering agents' randomness (Friedman & Mauersberger, 2022; Goeree et al., 2005; Sandomirskiy et al., 2024), people address other players' uncertainty in an integrated manner as a risk metric outside of their expected outcomes (e.g., $\mathbf{Risk}(\mathbb{E}_{\boldsymbol{a}_{-i} \in \pi_{-i}}[V_{i,h}^{\pi, P}(a_i, \boldsymbol{a}_{-i})])$), instead of in a separate manner as an expectation of the risk metric over outcomes of each joint action (namely, $\mathbb{E}_{\boldsymbol{a}_{-i} \in \pi_{-i}}[\mathbf{Risk}(V_{i,h}^{\pi, P}(a_i, \boldsymbol{a}_{-i}))]$). Here, the former one—which is more realistic—corresponds to our fictitious uncertainty set, while the latter one corresponds

to the uncertainty sets with $(s, \boldsymbol{a})$-*rectangularity condition* (Ma et al., 2023; Blanchet et al., 2023; Shi et al., 2024b) studied in prior works. Hence, the proposed uncertainty set modeling is realistic and predictive of human decision-making behaviors from behavioral economics.

- *A natural adaptation from single-agent robust RL.* When agents follow some joint policy $\pi : \mathcal{S} \times [H] \mapsto \Delta(\mathcal{A})$, fixing other agents' policy $\pi_{-i}$, from the perspective of each individual agent $i$, RMGs with our proposed $(s, a_i)$-rectangularity condition will degrade to a single-agent robust RL problem with the widely used $(s, a_i)$-rectangularity condition in the single-agent literature (Iyengar, 2005; Zhou et al., 2021). Namely, from any agent $i$'s viewpoint, in a RMG, it deals with a "fictitious" player that can not only manipulate the environmental dynamics but also other players' policy $\pi_{-i}$.

## 3.2 Properties of RMGs with fictitious uncertainty set

Throughout the paper, we focus on the class of RMGs with the above proposed fictitious uncertainty sets, denoted as $\mathcal{RMG}_{\text{in}}$ and abbreviated as fictitious RMGs in the remaining of the paper. In this section, we present key facts about fictitious RMGs related to best-response policies, equilibria, and the corresponding one-step lookahead robust Bellman equations. The proofs can be found in the full version (Shi et al., 2024a).

First, we introduce the following lemma, which verifies the existence of a robust best-response policy that achieves the maximum robust value function (cf. (6)) in any $\mathcal{RMG}_{\text{in}}$.

**Lemma 3.2.** *For any $i \in [n]$, given $\pi_{-i} : \mathcal{S} \times [H] \mapsto \Delta(\mathcal{A}_i)$, there exists at least one policy $\widetilde{\pi}_i : \mathcal{S} \times [H] \to \Delta(\mathcal{A}_i)$ for the $i$-th agent that can simultaneously attain $V_{i,h}^{\widetilde{\pi}_i \times \pi_{-i}, \sigma_i}(s) = V_{i,h}^{\star, \pi_{-i}, \sigma_i}(s)$ for all $s \in \mathcal{S}$ and $h \in [H]$. We refer this policy as the* robust best-response policy.

**Existence of robust NE and robust CCE.** Fictitious RMGs can be viewed as hierarchical games with $n + nS \sum_{i=1}^n A_i$ agents. This includes the original $n$ agents and $n$ additional sets of $S \sum_{i=1}^n A_i$ independent adversaries, each determining the worst-case transitions for one agent over a state plus agent-wise-action pair. Considering the solution concepts — robust NE and robust CCE — introduced in Section 2.2, the following theorem verifies the existence of them for any fictitious RMGs using Kakutani's fixed-point theorem (Kakutani, 1941), focusing on robust NE firstly.

**Theorem 3.3** (Existence of robust NE). *For any $\mathcal{RMG}_{\text{in}} = \{\mathcal{S}, \{\mathcal{A}_i\}_{1 \leq i \leq n}, \{\mathcal{U}_\rho^{\sigma_i}(P^0, \cdot)\}_{1 \leq i \leq n}, r, H\}$ with an uncertainty set defined in Definition 3.1, there exists at least one robust NE.*

Analogous to standard Markov games, since {robust NE} ⊆ {robust CCE}, Theorem 3.3 indicates the existence of robust CCEs directly. The class of fictitious RMGs feature a robust counterpart of the Bellman equation — *robust Bellman equation*, which is detailed in the full version (Shi et al., 2024a).

# 4 Sample-Efficient Learning: Algorithm and Theory

In this section, we focus on designing sample-efficient algorithms for solving fictitious RMGs when agents need to collect data by interacting with the unknown shared environment in order to learn the equilibria. To proceed, we shall first specify the data collection mechanism and the divergence function for the uncertainty set. Then we propose a sample-efficient algorithm Robust-Q-FTRL that leverages tailored adaptive sampling strategy to break the curse of multiagency for solving fictitious RMGs.

## 4.1 Problem setting and goal

Recall that the uncertainty sets are constructed by specifying a divergence function $\rho$ and the uncertainty level to control its shape and size. In this work, we focus on using the TV distance as the divergence function $\rho$ for the uncertainty set, following Szita et al. (2003); Lee et al. (2021); Pan et al. (2023); Shi et al. (2023; 2024b), defined by

$$\forall P, P' \in \Delta(\mathcal{S}): \quad \rho_{\mathsf{TV}}(P, P') := \frac{1}{2} \|P - P'\|_1. \quad (13)$$

For convenience, throughout the paper, we abbreviate $\mathcal{U}^{\sigma_i}(\cdot) := \mathcal{U}^{\sigma_i}_{\rho_{\mathsf{TV}}}(\cdot)$ when there is no ambiguity.

**Data collection mechanism: a generative model.** We assume the agents interact with the environment through a generative model (simulator) (Kearns & Singh, 1999), which is a widely used sampling mechanism in both single-agent RL and MARL (Zhang et al., 2020b; Li et al., 2022). Specifically, at any time step $h$, we can collect an arbitrary number of independent samples from any state and joint action tuple $(s, \boldsymbol{a}) \in \mathcal{S} \times \mathcal{A}$, generated based on the true *nominal* transition kernel $P^0$: $s^i_{h,s,\boldsymbol{a}} \overset{i.i.d}{\sim} P^0_h(\cdot \mid s, \boldsymbol{a})$ for $i = 1, 2, \ldots$.

**Goal.** Consider any fictitious RMGs $\mathcal{RMG}_{\mathsf{in}} = \{\mathcal{S}, \{\mathcal{A}_i\}_{1 \le i \le n}, \{\mathcal{U}^{\sigma_i}(P^0, \cdot)\}_{1 \le i \le n}, r, H\}$. In practice, learning exact robust equilibria is computationally challenging and may not be necessary, instead in this work, we focus on finding an approximate robust CCE (defined in (8)). Namely, a distribution $\xi := \{\xi_h\}_{h \in [H]} : [H] \times \mathcal{S} \mapsto \Delta(\prod_{i \in [n]} \Delta(\mathcal{A}_i))$ is said to be an $\varepsilon$-*robust CCE* if

$$\mathsf{gap}_{\mathsf{CCE}}(\xi) := \max_{s \in \mathcal{S}, 1 \le i \le n} \left\{ \mathbb{E}_{\pi \sim \xi} \left[ V^{\star, \pi_{-i}, \sigma_i}_{i,1}(s) \right] \right.$$

$$\left. - \mathbb{E}_{\pi \sim \xi} \left[ V^{\pi, \sigma_i}_{i,1}(s) \right] \right\} \le \varepsilon. \quad (14)$$

Armed with a generative model of the nominal environment, the goal becomes learning a robust CCE using as few samples from the simulator as possible.

## 4.2 Algorithm design

With the sampling mechanism over a generative model in hand, we propose an algorithm called Robust-Q-FTRL to learn an $\varepsilon$-*robust CCE* in a sample-efficient manner. The complete procedure is summarized in Algorithm 2. Robust-Q-FTRL draws inspiration from Q-FTRL developed in the standard MG literature (Li et al., 2022), but empowers tailored designs for learning in fictitious RMGs to achieve a robust equilibrium and to tackle statistical challenges arising from agents' nonlinear worst-case objectives.

**Constructing the empirical model via $N$-sample estimation.** For each time step $h$, we denote $\pi^k_{i,h}$ as the current learning policy of the $i$-th agent before the beginning of the $k$-th iteration for any $k \in [K]$. And we denote the joint product policy as $\pi^k_h = (\pi^k_{1,h}, \cdots, \pi^k_{n,h})$. During each iteration $k$, for each agent $i \in [n]$, we require to generate $N$ independent samples from the generative model over each $(s, a_i) \in \mathcal{S} \times \mathcal{A}_i$ to obtain an empirical model, detailed in Algorithm 1. It includes an empirical reward function represented by $r^k_{i,h} \in \mathbb{R}^{SA_i}$ and transition kernels denoted by $P^k_{i,h} \in \mathbb{R}^{SA_i \times S}$. Note that different from standard MGs, we need to generate $N$ samples instead of 1 sample per iteration to handle the additional statistical challenges induced by the non-linear objective of agents ($N$ will be specified in Theorem 4.1).

**Estimating robust Q-function of the current policy $\pi^k_h$.** We denote $\widehat{V}_{i,h} \in \mathbb{R}^S$ as the estimation of the $i$-th agent's robust value function at time step $h$. For any agent $i$, with the empirical reward function $r^k_{i,h}$, empirical kernel $P^k_{i,h}$, and the estimated robust value function $\widehat{V}_{i,h+1}$ at the next step in hand, the robust Q-function $\{q^k_{i,h}\}$ of current policy $\pi^k_h$ can be estimated as: for all $(i, h, s, a_i) \in [n] \times [H] \times \mathcal{S} \times A_i$,

$$q^k_{i,h}(s, a_i) = r^k_{i,h}(s, a_i) + \inf_{\mathcal{P} \in \mathcal{U}^{\sigma_i}(P^k_{i,h,s,a_i})} \mathcal{P}\widehat{V}_{i,h+1}. \quad (15)$$

Unlike the linear function w.r.t. $P^k_{i,h}$ in standard MGs, (15) lacks a closed form and introduces an additional inner optimization problem. Solving (15) directly is computationally challenging due to the need to optimize over an $S$-dimensional probability simplex, with complexity growing exponentially with the state space size $S$. Fortunately, by applying strong duality, we can solve (15) equivalently via its dual problem with tractable computation (Iyengar, 2005):

$$q^k_{i,h}(s, a_i) = r^k_{i,h}(s, a_i) + \max_{\alpha \in [\min_s \widehat{V}_{i,h+1}(s), \max_s \widehat{V}_{i,h+1}(s)]}$$

$$\left\{ P_{i,h}^k \left[\widehat{V}_{i,h+1}\right]_\alpha - \sigma_i \left(\alpha - \min_{s'} \left[\widehat{V}_{i,h+1}\right]_\alpha (s')\right) \right\}, \quad (16)$$

where $[V]_\alpha$ denotes the clipped version of any vector $V \in \mathbb{R}^S$ determined by some level $\alpha \geq 0$, namely,

$$[V]_\alpha(s) := \begin{cases} \alpha, & \text{if } V(s) > \alpha, \\ V(s), & \text{otherwise.} \end{cases} \quad (17)$$

The above two modules are key components of Robust-Q-FTRL, serving for constructing nonlinear robust objectives in the online learning process and ensuring the desired statistical accuracy.

**Overall pipeline of Robust-Q-FTRL.** With these modules in place, we introduce Robust-Q-FTRL, which follows a similar online learning procedure as Q-FTRL for standard MGs (Li et al., 2022). The complete procedure is summarized in Algorithm 2. We denote $Q_{i,h}^k \in \mathbb{R}^{SA_i}$ as the estimated robust Q-function of the equilibrium for the $i$-th agent at the $k$-th iteration of time step $h$. To begin with, Robust-Q-FTRL initialize the robust value function, robust Q-function $\widehat{V}_{i,H+1}(s) = Q_{i,h}^0(s,a_i) = 0$, and the policy $\pi_{i,h}^1(a_i \,|\, s) = 1/A_i$ for all $(i,s) \in [n] \times \mathcal{S}$. Then subsequently from the final time step $h = H$ to $h = 1$, for each step $h$, a $K$ iterations online learning process will be executed. At each $k$-th iteration, given current policy $\pi_h^k$, as described above, an empirical model ($\{r_{i,h}^k\}_{i\in[n]}$ and $\{P_{i,h}^k\}_{i\in[n]}$) is constructed by $N$-sample estimation (cf. Algorithm 1). Then the robust Q-function $\{q_{i,h}^k\}_{i\in[n]}$ of the current policy $\pi_h^k$ is estimated by (16).

Now we are ready to specify the loss objective and proceed the online learning procedure. With the current one-step update $\{q_{i,h}^k\}$, we update the Q-estimate as $Q_{i,h}^k = (1 - \alpha_k)Q_{i,h}^{k-1} + \alpha_k q_{i,h}^k$. Here, $\{\alpha_k\}_{k\in[K]}$ is a series of rescaled linear learning rates with some $c_\alpha \geq 24$, for all $k \in [K]$:

$$\alpha_k = \frac{c_\alpha \log K}{k - 1 + c_\alpha \log K}$$
$$\alpha_k^n = \begin{cases} \alpha_k \prod_{i=k+1}^n (1 - \alpha_i), & \text{if } 0 < k < n \leq K \\ \alpha_n & \text{if } k = n \end{cases} \quad (18)$$

Let the Q-estimate be the online learning loss objective at this moment, we apply the Follow-the-Regularized-Leader strategy (Shalev-Shwartz, 2012; Li et al., 2022) to update the corresponding policy as below:

$$\pi_{i,h}^{k+1}(a_i \,|\, s) = \frac{\exp\left(\eta_{k+1} Q_{i,h}^k(s,a_i)\right)}{\sum_{a'} \exp\left(\eta_{k+1} Q_{i,h}^k(s,a')\right)}$$
$$\text{with} \quad \eta_{k+1} = \sqrt{\frac{\log K}{\alpha_k H}}, \qquad k = 1, 2, \dots$$
$$(19)$$

This is a widely used adaptive sampling and learning procedure for MARL problems.

After completing $K$ iterations for time step $h$, we finalize the robust value function estimation by setting it to its confidence upper bound, incorporating carefully designed optimistic bonus terms $\{\beta_{i,h}\}$ as: for all $(i,h,s) \in [n] \times [H] \times \mathcal{S}$,

$$\beta_{i,h}(s) = c_{\mathsf{b}} \sqrt{\frac{\log^3\left(\frac{KS\sum_{i=1}^n A_i}{\delta}\right)}{KH}}$$
$$\sum_{k=1}^K \alpha_k^K \left\{ \mathsf{Var}_{\pi_{i,h}^k(\cdot|s)}\left(q_{i,h}^k(s,\cdot)\right) + H \right\}, \quad (20)$$

where $c_{\mathsf{b}}$ denotes some absolute constant, $\delta \in (0,1)$ is the high probability threshold, Finally, after the recursive learning process ends for all time steps $h = H, H-1, \cdots, 1$, we output a distribution of product policy $\widehat{\xi} = \{\widehat{\xi}_h\}_{h\in[H]}$ over all the policies $\{\pi_h^k = (\pi_{1,h}^k \times \cdots \times \pi_{n,h}^k)\}_{h\in[H],k\in[K]}$ occurs during the process that defined as

$$\forall (h,k) \in [H] \times [K]: \quad \xi_h(\pi_h^k) := \alpha_k. \quad (21)$$

### 4.3 Theoretical guarantees

In this section, we provide the theoretical guarantees for the sample complexity of our proposed algorithm Robust-Q-FTRL, shown as below:

**Theorem 4.1** (Upper bound). *Using the TV uncertainty set defined in* (13). *Consider any* $\delta \in (0,1)$ *and any fictitious RMGs* $\mathcal{RMG}_{\mathsf{in}} = \{\mathcal{S}, \{\mathcal{A}_i\}_{1\leq i\leq n}, \{\mathcal{U}^{\sigma_i}(P^0, \cdot)\}_{1\leq i\leq n}, r, H\}$ *with* $\sigma_i \in (0,1]$ *for all* $i \in [n]$. *For any* $\varepsilon \leq \sqrt{\min\left\{H, \frac{1}{\min_{1\leq i\leq n}\sigma_i}\right\}}$, *Algorithm* 2 *can output an* $\varepsilon$-*robust CCE* $\widehat{\xi}$, *i.e.,*

$$\mathsf{gap}_{\mathsf{CCE}}(\widehat{\xi}) \leq \varepsilon$$

*with probability at least* $1 - \delta$, *as long as*

$$N \geq \frac{C_1 H^2}{\epsilon^2} \min\left\{\frac{1}{\min_{1\leq i\leq n}\sigma_i}, H\right\}, \quad K \geq \frac{C_1 H^3}{\epsilon^2}. \quad (22)$$

*Here* $C_1$ *is some universal large enough constant. Namely, it is sufficient if the total number of samples acquired in the learning process obeys*

$$N_{\mathsf{all}} := HKNS \sum_{1\leq i\leq n} A_i$$
$$\geq \frac{(C_1)^2 H^6 S \sum_{1\leq i\leq n} A_i}{\varepsilon^4} \min\left\{H, \frac{1}{\min_{1\leq i\leq n}\sigma_i}\right\}.$$

Before we jump into more discussions of the above theorem, in addition, we introduce the information-theoretic minimax lower bound for this problem as well.

**Lower bound for learning in fictitious RMGs.** Considering the instances of fictitious RMGs that the action space for all the agents except the $i$-th agent contains only a single action, i.e., $A_j = 1$ for all $j \neq i$. As such, all the agents $j \neq i$ will take a fixed action and the game reduces to a single-agent robust MDP with $(s, a)$-*rectangularity condition* (Zhou et al., 2021). So the goal of finding the robust equilibrium — robust NE/CCE also degrades to finding the optimal policy of the $i$-th agent. Invoking the results from Shi et al. (2024b, Theorem 2), the lower bound for the class of fictitious RMGs is achieved directly: consider any tuple $\{S, \{A_i\}_{1 \leq i \leq n}, \{\sigma_i\}_{1 \leq i \leq n}, H\}$ obeying $\sigma_i \in (0, 1 - c_1]$ with $0 < c_1 \leq \frac{1}{4}$ being any small enough positive constant, and $H > 16 \log 2$. Let

$$\varepsilon \leq \begin{cases} \frac{c_1}{H}, & \text{if } \sigma_i \leq \frac{c_1}{2H}, \\ 1 & \text{otherwise} \end{cases} \quad (23)$$

We can construct a set of fictitious RMGs $\mathcal{M} = \{\mathcal{RMG}_{\mathsf{in}}^i\}_{i \in [I]}$, such that for any dataset generated from the nominal environment with in total $N_{\mathsf{all}}$ independent samples over all state-action pairs, we have $\inf_{\widehat{\xi}} \max_{\mathcal{RMG}_{\mathsf{in}}^i \in \mathcal{M}} \left\{ \mathbb{P}_{\mathcal{RMG}_{\mathsf{in}}^i}\left(\mathsf{gap}_{\mathsf{CCE}}(\widehat{\xi}) > \varepsilon\right) \right\} \geq \frac{1}{8}$ if

$$N_{\mathsf{all}} \leq \frac{C_2 S H^3 \max_{i \in [n]} A_i}{\varepsilon^2} \min\left\{ H, \frac{1}{\min_{i \in [n]} \sigma_i} \right\}. \quad (24)$$

Here, the infimum is taken over all estimators $\widehat{\xi}$, $\mathbb{P}_{\mathcal{RMG}_{\mathsf{in}}^i}$ denotes the probability when the game is $\mathcal{RMG}_{\mathsf{in}}^i$ for all $\mathcal{RMG}_{\mathsf{in}}^i \in \mathcal{M}$, and $C_2$ is some small enough constant.

Armed with both the upper bound (Theorem 4.1) and lower bound in (24), we are now ready to discuss the implications of our sample complexity results.

**Breaking the curse of multiagency in the sample complexity for RMGs.** Theorem 4.1 demonstrates that for any fictitious RMGs, Robust-Q-FTRL algorithm finds an $\epsilon$-robust CCE when the total number of samples exceeds

$$\widetilde{O}\left( \frac{SH^6 \sum_{1 \leq i \leq n} A_i}{\epsilon^4} \min\left\{ H, \frac{1}{\min_{1 \leq i \leq n} \sigma_i} \right\} \right).$$

To the best of our knowledge, Robust-Q-FTRL with the above sample complexity is the first algorithm for RMGs breaking the curse of multiagency, regardless of the types of uncertainty sets. Our sample complexity depends linearly on the sum of each agent's actions $\sum_{i=1}^n A_i$ rather than their product $\prod_{i=1}^n A_i$—making the algorithm highly scalable as the number of agents increases.

**Comparisons with prior works.** Prior works focus on learning equilibria for a different kind of robust MGs with $(s, \boldsymbol{a})$-rectangular uncertainty sets (Ma et al., 2023; Blanchet et al., 2023; Shi et al., 2024b). However, the state-of-the-art sample complexity $\widetilde{O}\left( \frac{SH^3 \prod_{i=1}^n A_i}{\varepsilon^2} \min\left\{ H, \frac{1}{\min_{1 \leq i \leq n} \sigma_i} \right\} \right)$ (Shi et al., 2024b) still suffers from the curse of multiagency with an exponential dependency on the number of agents when all agents have equal action spaces, which uses nonadaptive sampling. Our work circumvents the curse of multiagency by the introduction of a new class of fictitious RMGs inspired from behavioral economics, together with resorting to a tailored adaptive sampling and online learning procedure, providing a fresh perspective to learning practical-meaningful RMGs.

**Technical insights.** For sample complexity analysis, while previous works have addressed the curse of multiagency in sequential games like standard Markov games (MGs) and Markov potential games, these methods are not directly applicable to RMGs. Prior approaches assume a linear relationship between the value function and the transition kernel, allowing statistical errors across $K$ iterations to cancel out. However, in RMGs, the robust value function, due to its distributionally robust requirement, is highly nonlinear and often lacks a closed form, making it impossible to linearly aggregate statistical errors. To tackle the nonlinear challenges in RMGs, we design a variance-style bonus term through non-trivial decomposition and control of auxiliary statistical errors caused by nonlinearity, resulting in a tight upper bound on regret during the online learning process.

## 5 Conclusion

Robustness in MARL presents greater challenges than in single-agent RL due to the strategic interactions between agents in a game-theoretic setting. This work proposes a new class of RMGs with fictitious uncertainty sets that naturally extends from robust single-agent RL and addresses more realistic problems considering human features where each agent considers the uncertainty of others in an integrated manner. We then propose Robust-Q-FTRL, the first algorithm to break the curse of multiagency in RMGs regardless of the uncertainty set definitions, with sample complexity scaling polynomially with all key parameters. This opens up new research directions in MARL, such as uncertainty set selection and construction.

## Acknowledgements

The work of Y. Chi is supported in part by the grants NSF CCF-2106778 and CNS-2148212, and by funds from federal agency and industry partners as specified in the Resilient & Intelligent NextG Systems (RINGS) program. The work of L. Shi is supported in part by the Resnick Institute and Computing, Data, and Society Postdoctoral Fellowship at

California Institute of Technology. The work of E. Mazumdar is supported in part from NSF-2240110. The work of A. Wierman is supported in part from the NSF through CNS-2146814, CPS-2136197, CNS-2106403, NGSDI-2105648.

## Impact Statement

This paper presents work whose goal is to advance the field of game theory and its interaction with artificial intelligence. There are many potential societal consequences of our work, such as serving as reference for public policy and economics, none which we feel must be specifically highlighted here.

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

---

**Algorithm 1** $N$-sample estimation $\left(\pi_h = \{\pi_{j,h}\}_{j\in[n]}, i, h\right)$

---

1: **Initialization:** the reward $\widehat{r} = 0 \in \mathbb{R}^{SA_i}$ and the transition model $\widehat{P} = 0 \in \mathbb{R}^{SA_i \times S}$.
2: **for** $(s, a_i) \in \mathcal{S} \times \mathcal{A}_i$ **do**
3:     **for** $t = 1, 2, \cdots, N$ **do**
4:         Sample $\boldsymbol{a}^t(s, a_i) = [a_j(s, a_i)]_{1 \leq j \leq n}$ constructed by independent actions drawn from policy:

$$a_j(s, a_i) \stackrel{\text{ind.}}{\sim} \pi_{j,h}(\cdot \,|\, s) \quad (j \neq i) \qquad \text{and} \qquad a_i(s, a_i) = a_i. \tag{25}$$

5:         Sample from the generative model:

$$r_{i,h}^t(s, a_i) = r_{i,h}(s, \boldsymbol{a}^t(s, a_i)), \quad s_{s,a_i}^t \sim P_h\big(\cdot \,|\, s, \boldsymbol{a}^t(s, a_i)\big). \tag{26}$$

6:     **end for**
7:     Set $\widehat{r}(s, a_i) = \frac{1}{N}\sum_{t\in[N]} r_{i,h}^t(s, a_i)$ and $\widehat{P}(s' \,|\, s, a_i) = \frac{1}{N}\sum_{t\in[N]} \mathbb{1}\big\{s_{s,a_i}^t = s'\big\}$.
8: **end for**
9: **Return:** empirical model $(\widehat{r}, \widehat{P})$.

---

# A  Related works

**Breaking curse of multiagency for standard Markov games.** Breaking the curse of multiagency is a major and prevalent challenge in sequential games. In standard multi-agent general-sum MGs, it has been shown that learning a Nash equilibrium requires an exponential sample complexity (Song et al., 2021; Rubinstein, 2017; Bai & Jin, 2020). However, for other types of equilibria, such as CE and CCE, many works have successfully broken the curse of multiagency. Specifically, for finite-horizon general-sum MGs in the tabular setting with finite state and action spaces, Jin et al. (2021) developed the V-learning algorithm for learning CE and CCE with the sample complexity of $\widetilde{O}(H^6 S(\max_{i\in[n]} A_i)^2/\epsilon^2)$ and $\widetilde{O}(H^6 S \max_{i\in[n]} A_i/\epsilon^2)$, respectively; Daskalakis et al. (2023) achieved a sample complexity of $\widetilde{O}(H^{11} S^3 \max_{i\in[n]} A_i/\epsilon^3)$ for learning a CCE. Beyond tabular settings, Wang et al. (2023) and Cui et al. (2023) extended these results to linear function approximation, achieving sample complexities of $\widetilde{O}(d^4 H^6 \left(\max_{i\in[n]} A_i^5\right)/\epsilon^2)$ and $\widetilde{O}(H^{10} d^4 \log\left(\max_{i\in[n]} A_i\right)/\epsilon^4)$, respectively, where $d$ is the dimension of the linear features. For Markov potential games, a subclass of MGs, Song et al. (2021) provided a centralized algorithm that learns a NE with a sample complexity of $\widetilde{O}(H^4 S^2 \max_{i\in[n]} A_i/\epsilon^3)$.

**Finite-sample analysis for distributionally robust Markov games.** Robust Markov games under environmental uncertainty are largely underexplored, with only a few provable algorithms (Zhang et al., 2020a; Kardeş et al., 2011; Ma et al., 2023; Blanchet et al., 2023; Shi et al., 2024b). Existing sample complexity analyses all suffer from the daunting curse of multiagency issues, or impose an extremely restricted uncertainty level that can fail to deliver the desired robustness (Ma et al., 2023; Blanchet et al., 2024; Shi et al., 2024b). Specifically, they all consider a class of RMGs with the $(s, \boldsymbol{a})$-*rectangularity condition*, where the uncertainty sets for each agent can be decomposed into independent sets over each $(s, \boldsymbol{a})$ pair. Shi et al. (2024b) considered the generative model with an uncertainty set measured by the TV distance, Blanchet et al. (2023) treated a different sampling mechanism with offline data for both the TV distance and KL divergence. In addition, Ma et al. (2023) required the uncertainty level be much smaller than the accuracy-level and an instance-dependent parameter (i.e., $\sigma_i \leq \max\{\frac{\varepsilon}{SH^2}, \frac{p_{\min}}{H}\}$ for all $i \in [n]$). This can thus fail to maintain the desired robustness, especially when the accuracy requirement is high (i.e., $\varepsilon \to 0$) or the RMG has small minimal positive transition probabilities (i.e., $p_{\min} \to 0$).

**Robust MARL.** Standard MARL algorithms may overfit the training environment and could fail dramatically due to the perturbations and variability of both agents' behaviors and the shared environment, leading to performance drop and large deviation from the equilibrium. To address this, this work considers a robust variant of MARL adopting the distributionally robust optimization (DRO) framework that has primarily been investigated in supervised learning (Rahimian & Mehrotra, 2019; Gao, 2020; Bertsimas et al., 2018; Duchi & Namkoong, 2018; Blanchet & Murthy, 2019) and has attracted a lot of attention in promoting robustness in single-agent RL (Nilim & El Ghaoui, 2005; Iyengar, 2005; Badrinath & Kalathil, 2021; Zhou et al., 2021; Shi & Chi, 2024; Wang et al., 2024; Shi et al., 2023; Clavier et al., 2024). Beyond the RMG framework considered in this work, recent research has advanced the robustness of MARL algorithms from various perspectives, including resilience to uncertainties or attacks on states (Han et al., 2022; Zhou & Liu, 2023), the type of agents (Zhang et al., 2021), other agents' policies (Li et al., 2019; Kannan et al., 2023), offline data poisoning (Wu et al., 2024; McMahan

---

**Algorithm 2** Robust-Q-FTRL

---

1: **Input:** learning rates $\{\alpha_k\}$ and $\{\eta_{k+1}\}$, number of iterations $K$ per time step, and number of samples $N$ per iteration.
2: **Initialization:** $\widehat{V}_{i,H+1}(s) = Q_{i,h}^0(s, a_i) = 0$ and $\pi_{i,h}^1(a_i \,|\, s) = 1/A_i$ for all $i \in [n]$ and then all $(h, s, a_i) \in [H] \times \mathcal{S} \times \mathcal{A}_i$.
3: *// start recursive learning process.*
4: **for** $h = H, H - 1, \cdots, 1$ **do**
5:     **for** $k = 1, 2, \cdots, K$ **do**
6:         **for** $i = 1, 2, \cdots, n$ **do**
7:             *// construct empirical models and estimate current robust Q-function*
8:             $\left(r_{i,h}^k, P_{i,h}^k\right) \leftarrow N$-sample estimation $\left(\pi_h^k = \{\pi_{j,h}^k\}_{j \in [n]}, i, h\right)$. (Algorithm 1)
9:             Estimate the robust Q-function $q_{i,h}^k$ of current $\pi_h^k$ according to (16).
10:             *// online learning procedure*
11:             Update the Q-estimate $Q_{i,h}^k = (1 - \alpha_k)Q_{i,h}^{k-1} + \alpha_k q_{i,h}^k$ and apply FTRL:

$$\forall (s, a_i) \in \mathcal{S} \times \mathcal{A}_i : \quad \pi_{i,h}^{k+1}(a_i \,|\, s) = \frac{\exp\left(\eta_{k+1} Q_{i,h}^k(s, a_i)\right)}{\sum_{a'} \exp\left(\eta_{k+1} Q_{i,h}^k(s, a')\right)}. \tag{27}$$

12:         **end for**
13:     **end for**
14:     *// set the final robust value estimate at time step $h$.*
15:     **for** $i = 1, 2, \cdots, n$ **do**
16:         For all $s \in \mathcal{S}$: set $\beta_{i,h}(s)$ to be the optimistic bonus term in (20) and

$$\widehat{V}_{i,h}(s) = \min\left\{ \sum_{k=1}^K \alpha_k^K \left\langle \pi_{i,h}^k(\cdot \,|\, s), q_{i,h}^k(s, \cdot) \right\rangle + \beta_{i,h}(s),\ H - h + 1 \right\}. \tag{28}$$

17:     **end for**
18: **end for**
19: *Output:* a set of policies $\{\pi_h^k = (\pi_{1,h}^k \times \cdots \times \pi_{n,h}^k)\}_{k \in [K], h \in [H]}$ and a distribution $\widehat{\xi} = \{\widehat{\xi}_h\}_{h \in [H]}$ over them. For any time step $h$, $\widehat{\xi}_h$ is the distribution over $\{\pi_h^k\}_{k \in [K]}$ so that $\widehat{\xi}_h(\pi_h^k) = \alpha_k^K$.

---

et al., 2024), and nonstationary environment (Szita et al., 2003). A recent review can be found in Vial et al. (2022).

