# OpenReview forum: "Breaking the Curse of Multiagency in Robust Multi-Agent Reinforcement Learning"
_ICML.cc/2025/Conference — ICML 2025 poster_

### Official Review · Reviewer_Wjwo · 2025-03-07

**Overall Recommendation:** 3

**Summary:**

This paper studies the problem of multi-player general-sum robust Markov games (RMGs). Via proposing a new robustness measure called fictitious uncertainty set that centers at $(s,a_i)$ instead of $(s,\mathbf a)$, the authors break the curse of multi-agents (i.e., the dependency is $\sum_{i=1}^n \lvert A_i\rvert$ rather than $\prod_{i=1}^n \lvert A_i\rvert$) for the first time in RMGs.

**Claims And Evidence:**

There is no proof sketch in the main text, making it pretty hard to justify the main insight in Theorem 4. I trust the authors out of good faith.

**Essential References Not Discussed:**

I feel the discussion is pretty complete. See Questions part for several recent papers in (non-robust) linear MGs that I'm not sure whether relevant enough.

**Experimental Designs Or Analyses:**

N/A

**Methods And Evaluation Criteria:**

The models and metrics are standard and fair.

**Other Comments Or Suggestions:**

See below.

**Other Strengths And Weaknesses:**

As mentioned earlier, the presentation can be improved: There is too much background information, making the room for algorithms, discussions of technical contributions, and proof sketch very limited. Looks like the main text could be polished to make slightly more room (e.g., math equations in Theorem 4.1 are spaced in an luxury way), but I still suggest the authors to move more background information into the appendices for the ease of reading.

The convergence $O(\epsilon^{-4})$ is slow, but it's fine as it's the first to break curse of multi-agencies.

The assumption of generative model is also non-desirable, but it's also fine since it's also presented in previous works on RMG.

**Questions For Authors:**

In the recent literature of breaking the curse of multi-agents in Linear Markov Games (non-robust) ones, a similar idea of "independent linear approximation", which essentially assumes that Q-functions are linear as $Q(s,a_i)=\theta^T \phi(s,a_i)$ instead of the previous "global linear approximation" scheme that $Q(s,\mathbf a)=\theta^T \phi(s,\mathbf a)$. See below for a series of such paper, the first two are concurrent and the last two are concurrent:

1. Qiwen Cui, Kaiqing Zhang, and Simon Du. "Breaking the curse of multiagents in a large state space: Rl in markov games with independent linear function approximation." COLT 2023.
2. Yuanhao Wang, Qinghua Liu, Yu Bai, and Chi Jin. "Breaking the curse of multiagency: Provably efficient decentralized multi-agent rl with function approximation." COLT 2023.
3. Junyi Fan, Yuxuan Han, Jialin Zeng, Jian-Feng Cai, Yang Wang, Yang Xiang, and Jiheng Zhang. "Rl in markov games with independent function approximation: Improved sample complexity bound under the local access model." AISTATS 2024.
4. Yan Dai, Qiwen Cui, and Simon Du. "Refined sample complexity for markov games with independent linear function approximation." COLT 2024.

**Question**:
1. Is your idea of assuming structures around $(s,a_i)$ instead of $(s,\mathbf a)$ similar to the independent linear function approximation above?
2. Could you sketch how your model (compared to the previous $(s,\mathbf a)$ one) helps you to break the curse of multi-agency in a more technical way?

**Remark.** The current rating assumes all the proofs 1) are correct, and 2) do not contain significant technical innovations that can be of independent interest. Feel free to correct me if I am missing something.

**Relation To Broader Scientific Literature:**

The idea of fictitious uncertainty set looks interesting, but the authors didn't discuss a lot (and whether it can be useful somewhere else). Also, see the Questions for a series of recent similar ideas in (non-robust) Linear MGs; I'm not sure whether they're relevant enough, but they look a bit similar.

**Theoretical Claims:**

Didn't check the proof as there is no concise sketch in the main text or the appendices.

---

> ### Author Rebuttal · Authors · 2025-03-31
>
> ### Q1. The idea of fictitious uncertainty set looks interesting, can it be useful somewhere else.
>
> Fictitious uncertainty sets, inspired by behavioral economics, hold significant value in game theory and behavioral economics and have several meaningful applications:
>
> * **Understanding human preferences regarding risk and robustness under uncertainty**: Humans are generally risk-averse and prefer robustness under uncertainty [1](https://dl.acm.org/doi/10.1145/3490486.3538351) [2](https://arxiv.org/html/2502.11243v1) [3](https://link.springer.com/article/10.1007/s10683-005-5374-7), whose behaviors correspond to our fictitious uncertainty set, rather than the $(s,\mathbf{a})$-rectangular uncertainty set used in prior works. Applying our framework to real-world human data could help predict individual risk preferences and facilitate the design of personalized decision-making strategies for users.
> * **Improving robustness in safety-critical applications.** As mentioned in the introduction, numerous safety-critical scenarios can benefit from our proposed framework to identify optimal solutions for multi-agent interactions, such as financial markets, social dilemmas, autonomous driving, and human-robot interactions.
>
> ### 2. Further improve the presentation.
>
> We sincerely thank you for your valuable suggestion regarding the writing. In the revised version of our work, we have moved part of the background information to the Appendix and included the algorithm description in the main text.
>
> ### 3. The convergence rate
>
> We acknowledge that the $O(\epsilon^{-4})$ convergence rate may seem suboptimal but note that rate $O(\epsilon^{-4})$ or $O(\epsilon^{-3})$ are common in MARL literature, as seen in [4](https://arxiv.org/pdf/2302.03673) and [5](https://arxiv.org/pdf/2204.03991). As the first to break the curse of dimensionality in RMGs, we see great value in further improvements and plan to explore ways to accelerate the algorithm in future work.
>
> ### 4. The relationship of this work to linear MG works
> * Is the idea of assuming $(s,a_i)$ structures around instead of $(s, \textbf{a})$ similar to the independent linear function approximation in linear MGs?
>
>     **The proposed $(s,a_i)$ plays a fundamentally different role in robust MGs compared to the methods used in non-robust linear MGs..**
>
>     1. **Different problems and challenges: Nonlinearity in Robust MGs.** Robust MGs pose additional challenges compared to non-robust MGs due to the nonlinearity of the robust value function, unlike the linear payoff functions in non-robust MARL, which fundamentally alters the problem structure and solution methods.
>     2. **Inspired by game theory and behavioral economics.** The $(s,a_i)$ structure is considered for two key reasons: 1) Capturing human behavior in reality inspired by behavioral economics, **both $(s,a_i)$-rectangular structures and $s$-rectangular structures ($\otimes\_{s\in \mathcal{S}} \mathcal{U}\_\rho^{\sigma_i}(\mathbb{E}\_{\mathbf{a} \sim \pi_h} P\_{h, s, \mathbf{a}})$) can predict human behavior**, but $(s, \textbf{a})$ fails to do so; 2) We ultimately choose to consider the $(s,a_i)$ structure instead of the $s$-structure because, the $(s,a_i)$-rectangular set makes Nash (also CCE and CE) is guaranteed to exist, whereas this is not the case for the $s$-rectangular set.
>     3. **Distinct techniques for breaking the curse of dimensionality in $(s,a_i)$ structures** A direct indication is that our techniques developed for $(s,a_i)$ structures also apply to $(s)$-uncertainty set, whereas no corresponding result exists for non-robust linear MG.
>
>
> * How does your model technically break the curse of multi-agency compared to prior work?
>
>     A concise answer is that it allows for the decomposition of the non-linear payoff function in robust MGs. The key reason why the $(s, a_i)$ structure helps break the curse of dimensionality lies in its role within the Bellman equation:
>     $$
>     V^{\pi}\_{i,h}(s) = \mathbb{E}\_{\mathbf{a} \sim \pi_h(s)}[r\_{i,h}(s, \mathbf{a})] + \mathbb{E}\_{a_i \sim \pi_{i,h}(s)} \left[ \inf\_{\mathcal{U}\_{\rho}^{\sigma_i} \left(P^{\pi_{-i}}\_{h,s,a_i} \right)} P V^{\pi}\_{i,h+1} \right],
>     $$
>     $V^{\pi}\_{i,h}$ becomes a linear function with respect to the $i^{th}$ agent's policy $\pi_{i,h}$. This property also holds for the $(s)$-uncertainty set but not for the $(s, \mathbf{a})$ uncertainty set used in prior works. Consequently, we can leverage concentration inequalities to control the gap between the estimated and true value functions, effectively breaking the curse of dimensionality for $(s, a_i)$ and $(s)$ uncertainty sets, but not for $(s, \mathbf{a})$.

---

> > ### Comment · Reviewer_Wjwo · 2025-04-01
> >
> > Okay, I agree that while in both case $V_{i,h}^\pi(s)$ become linear in $\pi_{i,h}$, there aren't many similarities. Thank you for your clarification. I feel this paper is pretty interesting and recommend for an accept.

---

> > > ### Author Response · Authors · 2025-04-02
> > >
> > > Thank you so much to the reviewer for recognizing our insights and contribution and raising the score to support this work!

---

### Official Review · Reviewer_TP9Q · 2025-03-11

**Overall Recommendation:** 3

**Summary:**

The paper consider the problem of strategic interactions in uncertain environments; namely, robust Markov games (MGs). Robust Markov games are the multi-agent extension of Markov decision processes. The authors consider MGs where the transition kernel, i.e., the dynamics governing state transition, drift from some nominal value within a given uncertainty set; i.e., players might be trained on a game with a particular transition kernel, but when they deploy their policies, they do so on a game with slightly shifted transition kernel.

The goal of each agent is to compute policies that unilaterally perform well under the worst-case shift of the game's parameters (rewards and transitions). This objective gives rise to the notion of robust equilibria (robust Nash equilibrium, robust coarse-correlated equilibrium).

The authors contribute:
1) a new assumption on the uncertainty sets, *fictitious* uncertainty sets that depend on the s-and-action pair of each agent. They do not depend on the joint-action of all players
2) an algorithm with provable guarantees that converges to a robust coarse-correlated equilibrium in games that satisfy the latter assumption.

The sample and iteration complexity is polynomial in the natural parameters of the game and break the curse of multiagency. I.e., the dependence is on the sum of the size of individual action spaces and not the product.

**Claims And Evidence:**

In general, the authors clearly support their claims with formal proofs. One of the parts that is slightly confusing is when they note *"To the best of our knowledge, Robust-Q-FTRL with the above sample complexity is the first algorithm for RMGs breaking the curse of multiagency, regardless of the types of uncertainty sets."*

Yes, their algorithm breaks the curse of multiagency but this is due to the additional assumption. They convincingly argue that this is a reasonable assumption, nevertheless, the claim that the algorithm breaks the curse of multiagency in RMG is imprecise.

**Essential References Not Discussed:**

I do not know the area of robust MGs and MDPs very well so I do not know whether they are missing some crucial reference.

**Experimental Designs Or Analyses:**

I check the the validity of mathematical arguments.

**Methods And Evaluation Criteria:**

The methods used to support the claims where mathematical formal reasoning.

**Other Comments Or Suggestions:**

Equatoin 9: There is no description what a the cartesian product is over. I am guessing all $h,s,a$.

289 to 305: left col. notation $V_{i,h}^{\pi, P}(a_i, \bm{a}_{-i})$ is confusing. Why does the value function take actions as arguments?

306 to 317: is it not the case for $(s,\bm{a})$-rectangular to degrade to $(s,a)$-rectangularity condition in case other agents remain fixed?

**Other Strengths And Weaknesses:**

* One of the weakness of the writing. I struggled understanding their definitions and how their fictitious uncertainty sets compare with previous assumptions. It required opening the cited papers to understand that. Since the authors are introducing a new assumption, in my opinion, they should take the time to carry out an explicit comparison between their assumption and the state-joint-action rectangular uncertainty set assumption. (I.e., the most similar assumption to theirs.)

* There seems to be a gap in the understanding of the complexity of solving RMGs. Is it perhaps impossible to break it for state-joint-action rectangular uncertainty sets? Answering in the positive would strengthen the necessity of this paper assumption

* The claim that the curse of multiagency is surpassed is true but only thanks to this assumption.

**Questions For Authors:**

* Is it impossible to break the curse of multiagency for state-joint-action rectangular uncertainty sets?
* What are other assumption that can lead to polynomial sample complexity results and make sense?
* Would you be able to get better convergence rates if you used optimistic ftrl [Syrgkanis et al. 2015]?

Syrgkanis, V., Agarwal, A., Luo, H. and Schapire, R.E., 2015. Fast convergence of regularized learning in games. Advances in Neural Information Processing Systems.

**Relation To Broader Scientific Literature:**

The authors seem to cite most relevant work in MARL and robust MDPs and robust MGs. They even connect their assumptions to economic theory which I really appreciate as the assumption seems well-founded.

**Theoretical Claims:**

I went over the proof of theorem 3.1 in the appendix which proves the existence of robust NEs which also implies the existence of robust CCEs.

I also went the main parts of the proof of theorem 4.1 (main contribution of the paper) and believe that it is correct.

---

> ### Author Rebuttal · Authors · 2025-03-31
>
> ### 1. "The proposed algorithm is the first to overcome the curse of multiagency in RMGs, irrespective of the uncertainty set types." Is it due to an additional assumption?
> Our work breaks the curse of multiagency **through two key innovations: the introduction of a new class of fictitious RMGs and a new algorithm**. Both elements are essential. While the algorithm addresses a specific type of RMGs—the proposed fictitious RMGs—it is also the first to break the curse within the broader class of RMGs, as no prior work has achieved this, even for other subclasses of RMGs. Specifically,
> * **Novel and Realistic Assumption Inspired by Behavioral Economics** General RMGs are computationally intractable to solve, which leads to the widely used rectangularity assumptions. We didn't add addtional assumption, but replace the state-joint-action rectangular assumption used in prior work by a realistic rectangular assumption inspired by human behavior in behavioral economics. While this fictitious uncertainty set helps break the curse of multiagency, its primary purpose is to realistically model human behavior.
> * **Tailored Algorithm for Breaking the Curse of Multiagency.** Breaking the curse of multiagency in our proposed RMGs is more challenging than in standard MGs due to the nonlinearity of the robust value function, unlike the linear payoff functions in standard MARL. To address this, Algorithm 1 uses tailored sampling methods to handle nonlinearity and integrates them with a customized online learning algorithm.
>
>
> ### 2. Is it "impossible" to break the curse for state-joint-action rectangular sets?
> * **Possibly Impossible, but a decisive conclusion for this is hard.** We agree with the reviewer's intuition that it may be impossible. Prior work shows an exponential sample complexity lower bound for computing Nash in general-sum games [2](http://arxiv.org/abs/1606.04550). However, no work has established a lower bound for computing CCE/CE in any type of game. This remains a very open question and area in game theory and a promising direction for future research.
> * We emphasize that the proposed fictitious uncertainty set is inspired by how human behavior is observed in reality through behavioral economics, not because the state-joint-action approach from prior works is infeasible, prompting a simpler alternative.
>
> ### 3. Can other assumption lead to polynomial sample complexity results and make sense?
> A "reasonable" uncertainty set should ensure the well-posedness of the problem --- guaranteeing the existence of equilibria (e.g., Nash). This presents challenges in problem formulation (assumptions), requiring both a well-posed uncertainty set and feasible algorithms with polynomial sample complexity to compute the corresponding equilibria, which may involve trade-offs between the two. For example, $s$-rectangular uncertainty sets may ensure polynomial sample complexity but fail to guarantee Nash equilibrium, making them less desirable. Exploring other uncertainty sets that ensure well-posedness and allow tractable algorithm presents exciting opportunities for both game theory and MARL community.
>
> ### 4. Get better convergence rates if use optimistic FTRL [Syrgkanis et al. 2015]?
> A brief answer, based on the authors' intuition, is no. The non-linear payoff functions in RMGs create a dilemma between the statistical complexity of estimating the transition kernel and the regret from online adversarial learning algorithms (e.g., FTRL and optimistic FTRL). The current bottleneck lies in the statistical complexity of estimating the transition kernel. Since we already use a state-of-the-art online learning algorithm with optimal sample complexity in standard MARL [1](http://arxiv.org/abs/2208.10458), switching to optimistic FTRL is unlikely to improve the results further.
>
>
> ### 5. Others:
> * Adding detailed comparison between the proposed set and prior works.
>     * As the reviewer suggested, we will certainly include a more detailed comparison in the appendix of the revised version.
> * The reviewer is correct that the cartesian product in Equation (9) is over all $h,s,a_i$ for any $i$-th agent.
> * 289 to 305: Why does the value function take actions as arguments?
>     * Here, $V$ represents a general payoff function that naturally depends on all agents' actions, not the value function. We will revise $V$ to a different notation, $u$, to avoid ambiguity.
> * 306 to 317: Will $(s,\textbf{a})$-rectangular degrade to single-agent $(s,a)$-rectangular RMDP in case other agents remain fixed?
>     * No, it does not degrade to $(s,a)$-rectangular RMDPs. Since even if other agents' policies $\pi_{-i}$ are fixed but **stochastic**, their selected joint actions $\textbf{a}\_{-i}$ will vary, leading to corresponding uncertainty sets around each joint action $\textbf{a}\_{-i}$. As a result, other agents cannot be treated as a fixed part of the environment, preventing the model from simplifying to the single-agent $(s,a)$-rectangular case.

---

### Official Review · Reviewer_mEkj · 2025-03-15

**Overall Recommendation:** 3

**Summary:**

The paper proposes a robust multi-agent reinforcement learning framework based on a new fictitious uncertainty set. It proves the existence of robust Nash equilibria and coarse correlated equilibria then introduce a novel algorithm, Robust-Q-FTRL, which adaptively samples from a nominal generative model and solves a dual-optimization problem to estimate robust Q-values. Robust-Q-FTRL breaks the curse of multiagency and thus improves scalability compared to prior methods.

**Claims And Evidence:**

The paper’s claims that it breaks the curse of multiagency is supported by sample complexity guarantees.

**Essential References Not Discussed:**

No

**Experimental Designs Or Analyses:**

There are no experiments.

**Methods And Evaluation Criteria:**

The paper is mainly theoretical and it makes sense to compare time complexity with those of the existing uncertainty-set-based baselines.

**Other Comments Or Suggestions:**

It would be better if the authors could empirically compare the convergence rate of Robust-Q-FTRL with the baselines.

**Other Strengths And Weaknesses:**

Strengths:

- Breaking the curse of multi-agency is an important problem in MARL.
- The sample complexity bounds surpass those of existing baselines.

Weakness:

- The paper lacks empirical experiments, so the method’s practical performance remains uncertain.

**Questions For Authors:**

- Can some of the assumptions in Robust-Q-FTRL be relaxed to make it applicable to deep MARL?

**Relation To Broader Scientific Literature:**

Breaking the curse of multi-agency is a longstanding challenge in multi-agent RL, where exponential blow-up of joint actions severely hinders scalability. This paper aligns with recent theoretical efforts to provide polynomial sample complexity for robust MARL. It contributes to ongoing work on designing algorithms that offers strong theoretical guarantees of efficiency.

**Theoretical Claims:**

I did not do a detailed verification of the proofs for the theoretical claims. However, the arguments presented appear logically sound based on the provided explanations.

---

> ### Author Rebuttal · Authors · 2025-03-31
>
> We sincerely thank the reviewer for the careful reading of the paper and the insightful and valuable feedback.
>
> ### 1. Additional experiments verifying the effectiveness of Robust-Q-FTRL against baseline methods could be beneficial.
>
> Thank you very much for this valuable suggestion! As the reviewer rightly observed, in this work, we focus on taking an initial step toward developing a clear and realistic formulation and framework for players in multi-agent systems under uncertainty. Towards this, we introduce a fictitious uncertainty set inspired by behavioral economics, establish the existence of Nash equilibria (as well as CCE and CE) and propose an algorithm with theoretical guarantees. As the reviewer suggested, further experimental validation of both the proposed formulation and the proposed algorithm with comparison to baseline across diverse application scenarios would be highly valuable. In the future, we are considering several such scenarios, including autonomous driving simulations in CARLA or real-world experiements for human-robot interactions with the humanoid robot Unitree G1.
>
> ### 2. Can some of the assumptions in Robust-Q-FTRL be relaxed to make it applicable to deep MARL?
>
> Thank you for raising this valuable point! Both assumptions underlying Robust-Q-FTRL can be relaxed to better align with practical deep MARL settings, which are truly interesting future directions. Specifically:
> * **Relaxation of data collection through a generative model:** Our theoretical findings have the potential to extend to more practical data collection scenarios commonly used in deep MARL, such as online or offline settings. Exploring these extensions represents an interesting direction, which introduces additional challenges—particularly in online settings, which inherently pose difficulties in terms of statistical estimation and sample efficiency [2].
> * **Relaxation of the tabular Markov game formulation for MARL problems:** We believe our current results in tabular cases provide a strong foundation for exploring more general scenarios involving function approximation. This aligns closely with deep MARL, where neural networks are typically employed to approximate policies and Q/V-value functions. Nevertheless, adapting our approach to general robust MARL problems (e.g., robust linear MARL) will require distinct problem formulations—an open research area with no existing solutions to the best of our knowledge. Moreover, algorithm design and theoretical analysis frameworks would necessitate different assumptions, such as linearization conditions for linear function approximation [1] and realizability or low-rank structural assumptions for general function approximation.
>
> > [1] Yuanhao Wang, Qinghua Liu, Yu Bai, and Chi Jin. "Breaking the curse of multiagency: Provably efficient decentralized multi-agent rl with function approximation." COLT 2023. \
>     [2] Lu, Miao, et al. "Distributionally robust reinforcement learning with interactive data collection: Fundamental hardness and near-optimal algorithm." arXiv preprint arXiv:2404.03578 (2024).

---

### Official Review · Reviewer_5hU2 · 2025-03-17

**Overall Recommendation:** 4

**Summary:**

This paper addresses the robustness issue in MARL by proposing a novel approach based on fictitious uncertainty sets. The main contributions are as follows:
1. The authors define a new type of uncertainty set, which incorporates both environmental uncertainty and the behavior of other agents. Then they prove the existence of robust NE and CCE under it.
2. They also design a sample-efficient algorithm, Robust-Q-FTRL. The algorithm leverages tailored adaptive sampling strategy to find an approximate robust CCE, just with polynomial sample complexity.

**Claims And Evidence:**

All claims made in the submission are well-supported by clear and convincing evidence.

**Essential References Not Discussed:**

No other works related.

**Experimental Designs Or Analyses:**

There is no experiment, but comprehensive theoretical proofs, including the definition of the fictitious uncertainty set, the existence of robust NE and CCE, and the sample complexity analysis of the Robust-Q-FTRL.

**Methods And Evaluation Criteria:**

This paper uses sample complexity as its evaluation criteria to describe the effectiveness of the algorithm, and it’s meaningful.

**Other Comments Or Suggestions:**

In Section 1.2, the sentence "... through rational deviations.." (page 2, line 97) appears to have an extra period at the end, which should be removed for consistency.

**Other Strengths And Weaknesses:**

Strengths:
1. Inspired by behavioral economics, the proposed fictitious uncertainty set integrates environmental uncertainty and the behavior of other agents, better reflecting real-world human decision-making, demonstrating significant innovation.
2. The authors provide rigorous proofs for the existence of robust NE and CCE, along with detailed sample complexity analysis, offering solid theoretical support.
3. The Robust-Q-FTRL is the first algorithm with sample complexity scaling polynomially with all key parameters in RMGs, significantly improving scalability. This may open up new research directions in related area.
4. The paper is well-organized, with a clear presentation of the research background and preliminary knowledge. And the logical flow is rigorous, making it friendly to reader and easy to follow.
Weaknesses:
1. The paper lacks experiments or simulations to verify the effectiveness of the algorithm. While the theoretical contributions are notable, the absence of empirical evidence makes it difficult to fully assess its practical applicability.
2. There is no detailed discussion on computational complexity. Although the sample complexity is optimized, the algorithm's computational complexity may be still high, especially in large state and action spaces.

**Questions For Authors:**

I believe this is an excellent paper and I have only one question:
Q1: How does this work perform in real-world MARL tasks? Are there any practical applications that have already been implemented?
This question does not affect my overall evaluation of the paper.

**Relation To Broader Scientific Literature:**

It can open up new research directions in MARL, such as uncertainty set selection and construction.

**Theoretical Claims:**

No obvious errors were found, but further validation is still required.

---

> ### Author Rebuttal · Authors · 2025-03-31
>
> We sincerely thank the reviewer for recognizing and appreciating our contributions, both in terms of the problem formulation and the technical results. This acknowledgment is extremely rewarding!
>
> ### 1. How does this work perform in real-world MARL tasks, and are there existing practical applications?
>
> Thanks for pointing out this essential point! Definitely, practical usefulness is the main motivation behind our problem formulation and the proposed fictitious uncertainty set. Currently, two promising classes of applications emerge from this work, providing exciting future research directions:
> * **Understanding human preferences regarding risk and robustness under uncertainty**: A classical finding in behavioral economics is that humans are typically risk-averse and prefer robustness when facing uncertainty stemming from other players or the environment [1]. Applying our algorithms to real-world human data could help predict individual risk preferences and facilitate the design of personalized decision-making strategies for users.
> * **Improving robustness in safety-critical applications.** As mentioned in the introduction, numerous safety-critical scenarios can benefit from our proposed framework to identify optimal solutions for multi-agent interactions, such as financial markets, social dilemmas, autonomous driving, and human-robot interactions. Various experimental scenarios include  autonomous driving simulations in CARLA, financial analysis using Kaggle datasets, and other datasets in Hugging Face, as well as real-world experiements for human-robot interactions with the humanoid robot Unitree G1.
>
> In this work, we focus on taking an initial step toward developing a clear and realistic formulation and framework for players in multi-agent systems. As the reviewer suggested and recognized, the next step is to apply this framework to diverse practical scenarios.
>
> ### 2. Further experiments to verify the effectiveness of the algorithm.
>
> Thank you very much for this valuable suggestion! In this work, we focus on taking an initial step toward developing a clear and realistic formulation and framework for players in multi-agent systems. As the reviewer suggested, further experimental validation of both the proposed problem formulation and the corresponding algorithms would be highly valuable across diverse application scenarios. In the future, we are considering several such scenarios, including autonomous driving simulations in CARLA or real-world experiements for human-robot interactions with the humanoid robot Unitree G1.
>
> ### 3. Discussions on the computational complexity of Robust-Q-FTRL
>
> We sincerely thank the reviewer for this valuable suggestion. In summary, the computational complexity of our proposed Robust-Q-FTRL algorithm is similar to that of the current state-of-the art algorithm for standard MARL presented in [2]. Specifically, Robust-Q-FTRL converges within $K=O(\frac{H^3}{\epsilon^2})$ iterations, with each iteration requiring computational complexity $O(HS\log(S)\sum_{i}A_i)$, which is nearly linear with respect to the size of the state and action spaces. As the reviewer suggested, we will incorporate this detailed discussion following the introduction of our main result, Theorem 4.1.
>
>
> ### 4. An extra period at the end of page 2, line 97
> Thank you for pointing this out. We have removed the extra period and will thoroughly polish the entire manuscript again in the revised version.
>
> > [1] Goeree, Jacob K., Charles A. Holt, and Thomas R. Palfrey. "Risk averse behavior in generalized matching pennies games." Games and Economic Behavior 45.1 (2003): 97-113. \
>     [2] Li, Gen, et al. "Minimax-optimal multi-agent RL in Markov games with a generative model." Advances in Neural Information Processing Systems 35 (2022): 15353-15367.

---

### Decision · Program_Chairs · 2025-05-01

**Decision:**

Accept (poster)

**Comment:**

All reviewers agree that the paper has an interesting contribution and should be accepted.